# Spatial clustering of CD68+ tumor associated macrophages with tumor cells is associated with worse overall survival in metastatic clear cell renal cell carcinoma

Nicholas H. Chakiryan[1]*, Gregory J. Kimmel[2], Youngchul Kim[3], Ali Hajiran[1], Ahmet M. Aydin[1], Logan Zemp[1], Esther Katende[1], Jonathan Nguyen[4], Neale Lopez-Blanco[4], Jad Chahoud[1], Philippe E. Spiess[1], Michelle Fournier[1], Jasreman Dhillon[4], Liang Wang[5], Carlos Moran-Segura[4], Asmaa El-Kenawi[6], James Mulé[6], Philipp M. Altrock[2], Brandon J. Manley[1]

**1** Department of Genitourinary Oncology, H Lee Moffitt Cancer Center and Research Institute, Tampa, Florida, United States of America, **2** Integrated Mathematical Oncology Department, H Lee Moffitt Cancer Center and Research Institute, Tampa, Florida, United States of America, **3** Department of Biostatistics and Bioinformatics, H Lee Moffitt Cancer Center and Research Institute, Tampa, Florida, United States of America, **4** Department of Pathology, H Lee Moffitt Cancer Center, Tampa, Florida, United States of America, **5** Department of Tumor Biology, H Lee Moffitt Cancer Center and Research Institute, Tampa, Florida, United States of America, **6** Immunology Department, H Lee Moffitt Cancer Center and Research Institute, Tampa, Florida, United States of America

* Nicholas.chakiryan@moffitt.org

**Data Availability Statement:** All relevant data are within the manuscript and its Supporting information files.

## Abstract

Immune infiltration is typically quantified using cellular density, not accounting for cellular clustering. Tumor-associated macrophages (TAM) activate oncogenic signaling through paracrine interactions with tumor cells, which may be better reflected by local cellular clustering than global density metrics. Using multiplex immunohistochemistry and digital pathologic analysis we quantified cellular density and cellular clustering for myeloid cell markers in 129 regions of interest from 55 samples from 35 patients with metastatic ccRCC. CD68+ cells were found to be clustered with tumor cells and dispersed from stromal cells, while CD163+ and CD206+ cells were found to be clustered with stromal cells and dispersed from tumor cells. CD68+ density was not associated with OS, while high tumor/CD68+ cell clustering was associated with significantly worse OS. These novel findings would not have been identified if immune infiltrate was assessed using cellular density alone, highlighting the importance of including spatial analysis in studies of immune cell infiltration of tumors.

**Significance**: Increased clustering of CD68+ TAMs and tumor cells was associated with worse overall survival for patients with metastatic ccRCC. This effect would not have been identified if immune infiltrate was assessed using cell density alone, highlighting the importance of including spatial analysis in studies of immune cell infiltration of tumors.

**Funding:** This work was supported by the Urology Care Foundation Research Scholar Award Program and Society for Urologic Oncology (to BJM); the United States Army Medical Research Acquisition Activity Department of Defense (KC180139 to BJM); Total Cancer Care Protocol at Moffitt Cancer Center, which was enabled in part by the generous support of the DeBartolo Family; the Biostatistics and Bioinformatics Shared Resource at the H. Lee Moffitt Cancer Center & Research Institute, a National Cancer Institute designated Comprehensive Cancer Center (P30-CA076292); and the Tissue Core Facility at the H. Lee Moffitt Cancer Center & Research Institute (P30-CA076292). The content is solely the responsibility of the authors and does not necessarily represent the official views of the American Urological Association or the Urology Care Foundation.

**Competing interests:** The corresponding author certifies that all conflicts of interest, including specific financial interests and relationships and affiliations relevant to the subject matter or materials discussed in the manuscript (ie. employment/affiliation, grants or funding, consultancies, honoraria, stock ownership or options, expert testimony, royalties, or patents filed, received, or pending), are the following: NHC, GJK, AH, AMA, LZ, JN, JC, SF, MF, JD, SM, CM, EK, PMA, and YK have no disclosures; BJM is an NCCN Kidney Cancer Panel Member; PES is an NCCN Bladder and Penile Cancer Panel Member and Vice-Chair; JM is an Associate Center Director at Moffitt Cancer Center, has ownership interest in Fulgent Genetics, Inc., Aleta Biotherapeutics, Inc., Cold Genesys, Inc., Myst Pharma, Inc., and Tailored Therapeutics, Inc., and is a consultant/advisory board member for ONCoPEP, Inc., Cold Genesys, Inc., Morphogenesis, Inc., Mersana Therapeutics, Inc., GammaDelta Therapeutics, Ltd., Myst Pharma, Inc., Tailored Therapeutics, Inc., Verseau Therapeutics, Inc., Iovance Biotherapeutics, Inc., Vault Pharma, Inc., Noble Life Sciences Partners, Fulgent Genetics, Inc., UbiVac, LLC, Vycellix, Inc., and Aleta Biotherapeutics, Inc. This does not alter our adherence to PLOS ONE policies on sharing data and materials.

## Introduction

Immune cell infiltration is typically quantified using cell density, which does not account for the local clustering of immune cells within the tumor-immune microenvironment (TIME). Localized cellular clustering of immune cells in the TIME is a seldom-used metric in studies of immune infiltration of tumors, but when applied has yielded novel and impactful findings across a variety of primary tumor sites and immune cell types [1–7].

Tumor-associated macrophages (TAM) activate oncogenic signaling and facilitate tumor growth and progression by secreting cytokines, growth factors, and angiogenic mediators, as well as a variety of proteases that activate additional growth factors and angiogenic mediators embedded in the extracellular matrix [7–14]. These paracrine interactions are concentration-gradient-dependent, and as such, the underlying biology may be better reflected by measures of local cellular clustering as opposed to global density metrics. Prior work has demonstrated an association between TAM density and worse overall survival (OS) in patients with clear cell renal cell carcinoma (ccRCC), but none have assessed TAM cellular clustering [9, 12, 14, 15].

Our primary objective was to determine whether cellular clustering of TAMs and tumor cells at the tumor/stromal interface was associated with worse OS for patients with metastatic ccRCC. The secondary objective was to describe the relative affinity for TAMs to be located in either the tumor or stromal compartments within the TIME.

## Methods

### Patient selection and specimen collection

We obtained 55 primary and metastatic tumor samples from 35 patients with metastatic ccRCC. For patients with multiple samples (N = 13), only the primary tumor sample was considered for the survival analysis. This study was reviewed and approved by the Advarra institutional review board (H. Lee Moffitt Cancer Center and Research Institute's Total Cancer Care protocol MCC# 14690; Advarra IRB Pro00014441). Written informed consents were obtained from all tissue donors. Patients were included in this study if they (1) were diagnosed with metastatic ccRCC; (2) provided written consent to analysis of their tissue; and (3) did not receive any systemic therapy prior to initial tissue collection.

### Multiplex immunofluorescent tissue staining

To prepare the tissue blocks, an experienced genitourinary pathologist (JD) reviewed each formalin fixed paraffin-embedded tissue sample and annotated 3 separate ROIs from the tumor-stroma-interface. The tumor-stroma-interface ROIs were selected such that each ROI contained approximately 50% tumor cells and 50% adjacent stroma, as to determine the relative affinity for myeloid cells to cluster into the tumor or stroma compartment. Tissue samples were then stained using the PerkinElmer OPAL 7 Color Automation Immunohistochemistry Kit (PerkinElmer, Waltham, MA) on the BOND RX Autostainer (Leica Biosystems, Vista, CA). In brief, tissue slides were sequentially stained using antibodies targeting CD68, CD163, and CD206. These markers were selected for their previously demonstrated frequency and impact in TAM studies in ccRCC. All subsequent steps, including deparaffinization, antigen retrieval, and staining, were performed using the OPAL manufacturer's protocol. Pan-cytokeratin and 4′,6-diamidino-2-phenylindole (DAPI) counterstaining were applied to all slides, and imaging was performed using the Vectra3 Automated Quantitative Pathology Imaging System (PerkinElmer, Waltham, MA).

## Quantitative image analysis

Multi-layer TIFF images were exported from InForm (PerkinElmer) and loaded into HALO 121 (Indica Labs, New Mexico) for quantitative image analyses. The size of the ROIs was standardized at $1356 \times 1012$ pixels, with a resolution of 0.5 µm/pixel, for a total surface area of 0.343 mm$^2$. For each staining marker, a positivity threshold within the nucleus or cytoplasm was set by an experienced digital image analysist (JN), and the entire image set was analyzed. The generated data included the total cell count, positive cell counts of each IF marker, fluorescence intensity of every individual cell, Cartesian coordinates for each cell, and the percent of cells that were positive for each marker.

## Spatial analysis

Cellular density was calculated globally for each ROI and defined as the number of cells per mm$^2$. ROIs containing $\geq 10$ cells positive for a relevant marker were considered eligible for spatial analysis. As there is no previously validated standard for this cutoff, the $\geq 10$ cell cutoff was agreed upon through consensus of the authors. Cellular clustering was quantified using the Ripley's K function, a methodology for quantifying spatial heterogeneity most commonly utilized in ecology, with isotropic edge correction, with the following normalization applied: $nK(r) = K(r) / \pi r^2$, as described previously [16–18] (Fig 1). As such, the expected value of nK(r) for all radii is 1.0, with values >1.0 representing cellular clustering, and values <1.0 representing cellular dispersion. The range of possible values for nK(r) is 0 to infinity. The nK(r) value is an observed over expected ratio (i.e. Tumor/CD68+ nK(25) = 1.30 can be interpreted as: "There were 30% more CD68+ cells observed within a 25um radius of each tumor cell than would be expected if the cells were randomly distributed."). To reflect cellular clustering at a localized distance, nK(r) at a radius of 25um was utilized in this analysis and will henceforth be referred to as nK(25). The search-circle radius value of 25um was selected as it represents approximately double that of a typical ccRCC tumor cell radius, and as such should represent the area in the immediate vicinity of the cell. Examples of point pattern plots with high or low CD68+ density and tumor/CD68+ clustering, as measured by nK (25), are provided in Fig 2B.

## Statistical analysis

Spearman's correlation coefficient was determined for CD68+ cell density and CD68 +/tumor cell clustering, to determine the relationship between cell density and cell clustering. Pairwise nK(25) values were generated for each marker type as they related to tumor and stromal cells, for each ROI. These pairwise nK(25) values, per ROI, were compared within each marker type using Wilcoxon signed-rank testing. OS was defined using time from sample collection to death or censoring at last follow-up. Cell density and nK(25) cutpoints for cohort stratification were determined using optimal cut-point methodology, minimizing the log-rank p-value for OS, as previously described [19]. Multivariable Cox regression was used to determine associations with OS, using age and International Metastatic RCC Database Consortium (IMDC) risk score as covariates [20]. A post-hoc power analysis was conducted to assess the minimum detectable hazard ratio for the Cox model. Statistical significance was defined as two-tailed p<0.05. Statistical analyses were performed using R program version 4.0.2 (Vienna, Austria), and spatial analysis was performed using the "spatstat" package [21].

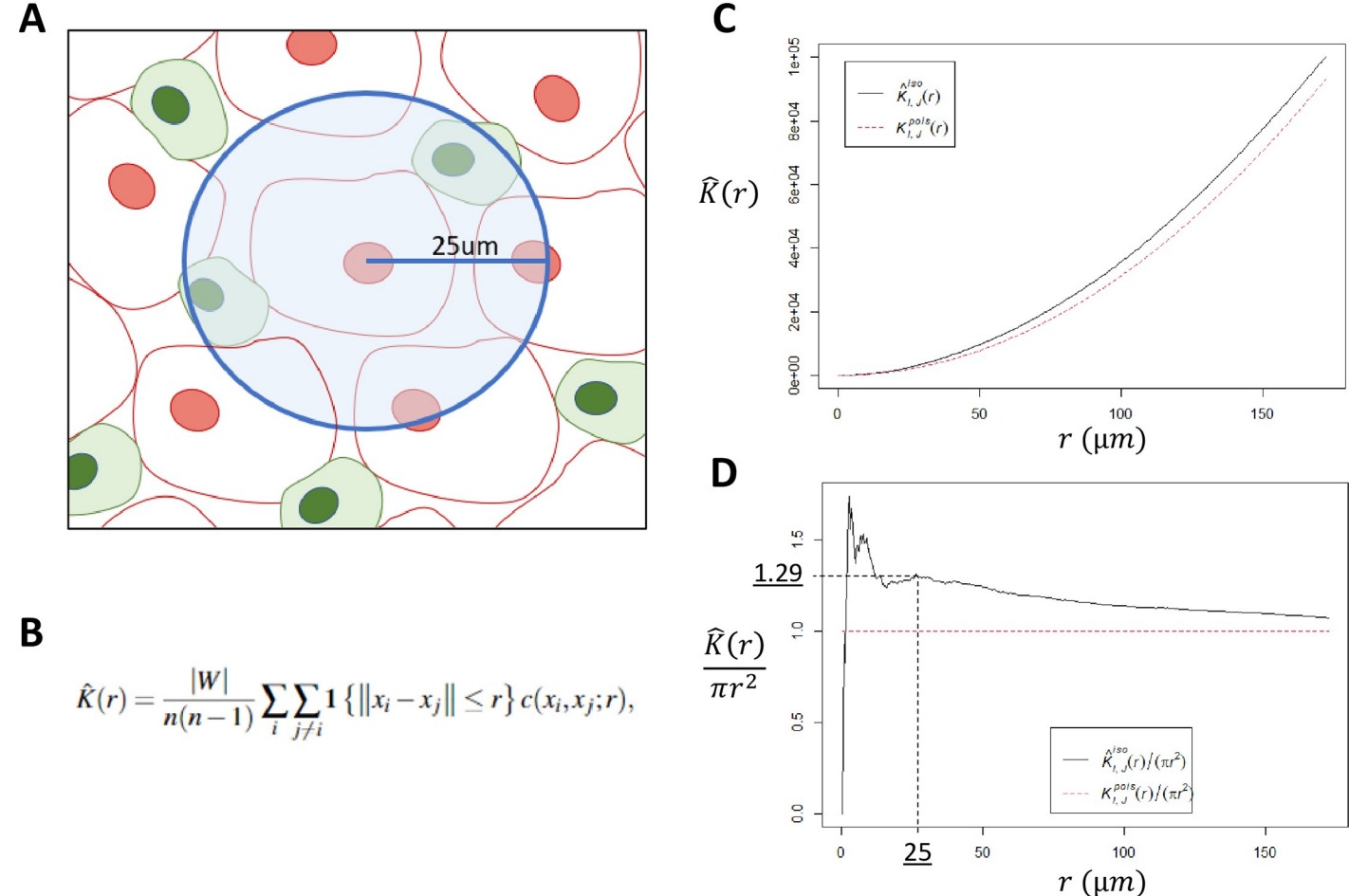

**Fig 1. Ripley's K function.** A: Illustrative representation of the pairwise Ripley's K function, where the number of cells of interest is identified within a search radius from another cell type, repeating the process over a continuum of radii, for each cell in the study area. B: The Ripley's K function estimator, where *n* is the total number of points in the study area, *W* is the study area, $1\{|xi\text{-}xj|\leq r\}$ is an indicator worth a value of 1 if points *i* and *j* are within distance *r*, and *c(xi,xj;r)* represents the applied edge correction (this analysis utilized isotropic edge correction). C: Plot of the naïve K(r) function using a representative slide from our analysis. The black solid line is the observed K(r), and dotted red line is the expected distribution if cells were randomly distributed, assuming a Poisson distribution. D: Normalization of the naïve Ripley's K function into $K(r)/\pi r^2$, resulting in an expected distribution of 1 for all values of *r*. The search radius utilized in this analysis was 25um. In this manuscript, K $(25um)/\pi r^2$ is abbreviated to "nK(25)".

## Results

### Patient characteristics

35 patients with metastatic ccRCC met criteria for inclusion. The median patient age was 58 (IQR 53–65), median primary tumor size was 8.0cm (IQR 6.0–10.3), 24 patients (69%) were male, 31 (89%) had a primary tumor grade of 3 or higher, and all patients were IMDC risk score 1 or greater (Table 1).

From the 55 samples obtained from this cohort, 129 ROIs were analyzed from the tumor/stroma interface.

### TAM cellular density and TAM/tumor cell clustering metrics are poorly correlated

Cellular density (cells/mm$^2$) and clustering (nK(25)) metrics were determined, as detailed in the methodology (Figs 1 and 2). The Spearman's correlation coefficient for CD68+ cell density

## A

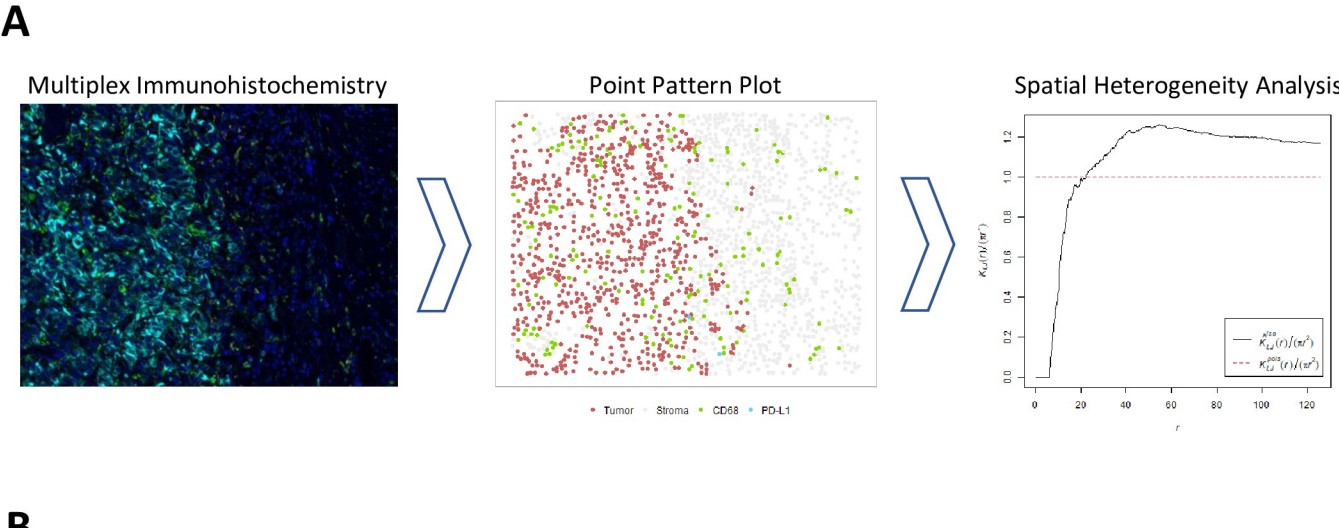

## B

### CD68+ Cell Density (cells/mm²)

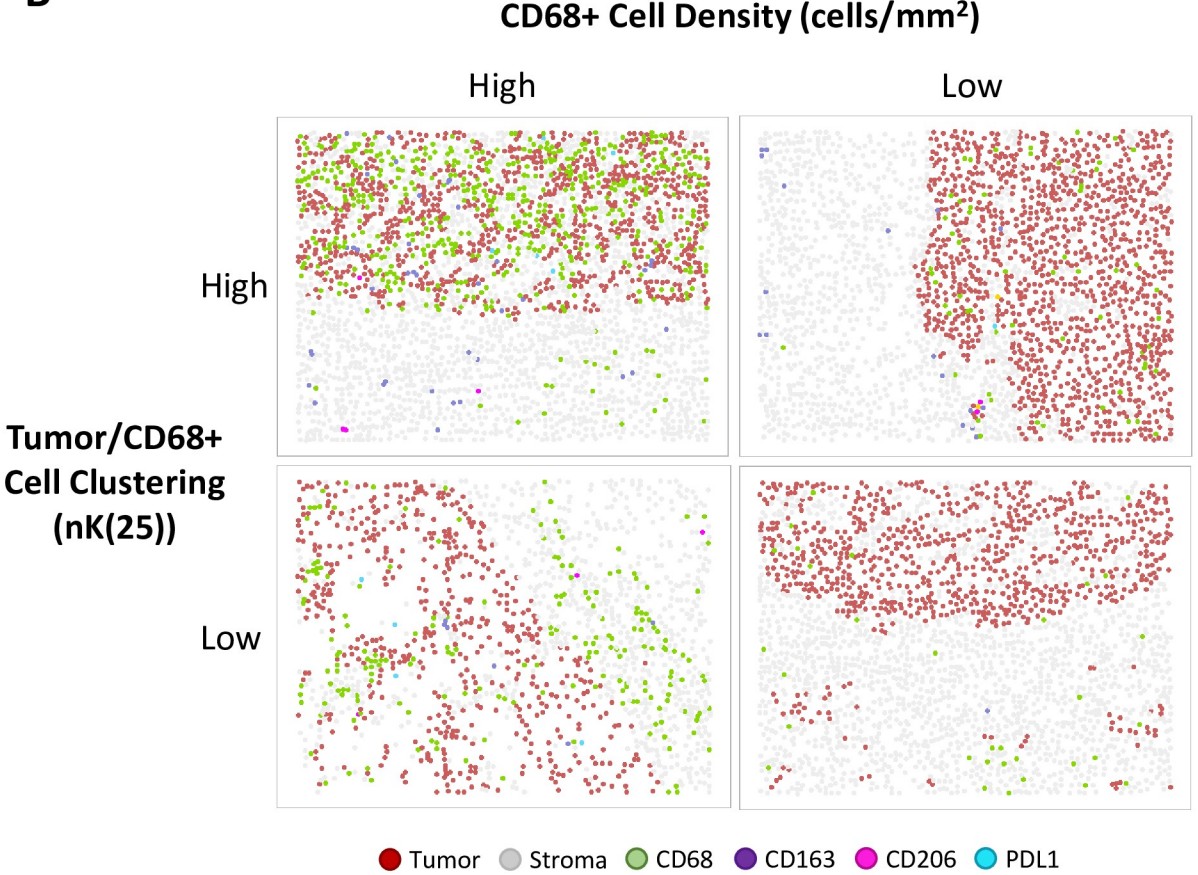

**Fig 2.** A: Project workflow, in brief. IHC staining for myeloid markers (CD68, CD163, CD206) and PD-L1. Digital pathologic analysis is utilized to convert IHC slides to point pattern plots, which are then utilized to calculate measures of spatial heterogeneity. B: Examples of point pattern plots demonstrating high or low CD68+ cell density (cells/mm2; cut-point = 151.273), and high or low Tumor/CD68+ cell clustering (nK(25), cut-point = 1.30).

**Table 1. Baseline patient characteristics at the time of sample collection.**

| Characteristic | N = 35[1] |
|---|---|
| Age (yrs) | 58 (53, 65) |
| IMDC | |
| 1 | 13 (37%) |
| 2 | 17 (49%) |
| 3+ | 5 (14%) |
| Gender | |
| Female | 11 (31%) |
| Male | 24 (69%) |
| Tumor Grade | |
| 2 | 4 (11%) |
| 3 | 21 (60%) |
| 4 | 10 (29%) |
| Primary Tumor Size (cm) | 8.0 (6.0, 10.3) |

[1] Statistics presented: median (IQR); n (%)

and tumor/CD68+ nK(25) indicated a weak negative correlation (R = -0.19, p = 0.046). This finding confirms that the cellular density and clustering metrics are not redundant (Fig 3A) within our cohort.

## TAMs have distinct affinities for the tumor and stromal compartments based on marker type

CD68+ TAMs were found to be clustered with tumor cells and dispersed from stromal cells (nK(25) = 1.10 and 0.90, respectively, p<0.01). TAMs expressing M2-phenotype markers CD163+ or CD206+ were found to be dispersed from tumor cells, and clustered with stromal cells (CD163: nK(25) = 0.77 and 1.15, respectively, p<0.01; CD206: nK(25) = 0.70 and 1.17, respectively, p<0.01). PD-L1+ cells did not demonstrate statistically significant spatial differences regarding their clustering with tumor and stromal cells (nK(25) = 0.78 and 0.72, respectively, p = 0.4) (Fig 3B). These findings suggest that TAMs may have varying biologic affinity for the tumor and stromal compartments based on their polarization phenotype.

## High CD68+ TAM/tumor cellular clustering is associated with worse overall survival

Multivariable Cox regression analysis revealed that high CD68+ cell density was not associated with OS (HR = 1.68, 95%CI 0.48–22.8, p = 0.2), while high tumor/CD68+ clustering was associated with significantly worse OS (HR = 6.19, 95%CI 1.16–33.1, p = 0.033) (Fig 3C). After stratifying for both cell density and cell clustering, patients with high CD68+ density and high tumor/CD68+ clustering were found to have significantly worse OS (HR = 8.50, 95%CI 1.97–36.7, p = 0.004) (Fig 3D).

A post-hoc power analysis using a power of 0.80, alpha of 0.05, and event probability of 80% demonstrated that with our 35-patient cohort the Cox model would be adequate to detect a minimum detectable hazard ratio of 1.7 for a standardized continuous variable in a univariable Cox proportional hazards regression analysis.

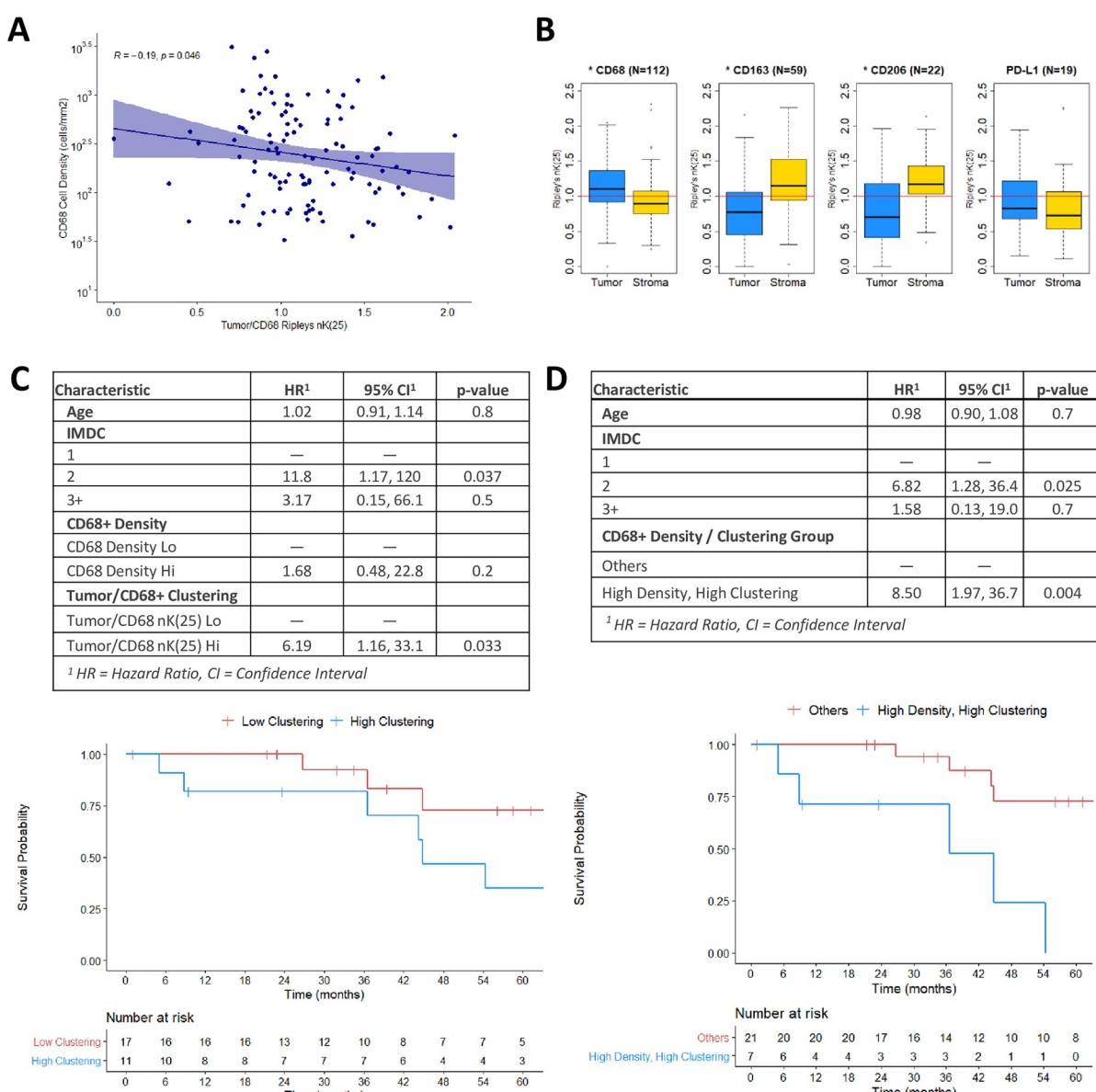

**Fig 3.** A: Scatter plot and Spearman's correlation of CD68+ cell density (cells/mm2) and Tumor/CD68+ (nK(25)), per ROI (N = 129, r = -0.19, p = 0.046). B: Boxplot diagrams of Ripley's nK(25) values for Tumor/CD68+, Stroma/CD68+, Tumor/CD163+, Stroma/CD163+, Tumor/CD206+, Stroma/CD206+, Tumor/PDL1+, and Stroma/PDL1+, for all ROIs (N = 129). nK(25) values > 1 indicate cellular clustering, and values <1 indicate cellular dispersion. Asterix in plot title denotes Wilcoxon test p<0.05. C: Multivariable Cox regression for OS, using CD68+ cellular density and Tumor/CD68+ nK(25) as separate covariates, with associated KM estimates for high versus low Tumor/CD68 + nK(25). D: Multivariable Cox regression for OS, stratifying patients who had high CD68+ cellular density and high Tumor/CD68+ nK(25) versus other patients, with associated KM estimates.

## High CD163+ TAM density is associated with worse overall survival

A robust survival analysis using CD163+ and CD206+ cell clustering was precluded by low populations of patients eligible for spatial analysis using these markers (N = 10 and N = 2, respectively) due to low cell counts. Multivariable Cox regression using CD163+ density, excluding clustering, revealed worse OS for patients with high CD163+ density (HR 19.4, 95%

CI 3.6–105.0, p<0.001). CD206+ cell density was not associated with OS (HR 2.21, 95%CI 0.6–8.0, p = 0.2) (S1 Fig).

## Discussion

Primarily, this analysis demonstrates the importance of performing spatial analysis when conducting investigations into immune cell infiltration of tumors. In this cohort, stratification by cellular density of CD68+ TAMs alone was insufficient to identify a survival difference. The addition of tumor/CD68+ cellular clustering was necessary to elucidate the clinical impact of CD68+ TAM infiltration. Furthermore, when CD68+ cell density and tumor/CD68+ cellular clustering were included as covariates in the same Cox regression, tumor/CD68+ clustering had a stronger association with OS (Fig 3C).

Measuring cellular clustering at a local level (25um) is a logical approach for assessing TAM infiltration, as their impact on tumor biology is via concentration-gradient-dependent effects occurring at close proximity. Thus, a localized clustering metric such as nK(25) would be expected to reflect the underlying biology of this interaction more so than a global metric such as cell density. To our knowledge, this is the first analysis to investigate the clinical impact of TAM clustering in ccRCC.

Secondarily, this analysis confirms previous work demonstrating the negative prognostic impact of TAM infiltration in ccRCC patients. It has been suggested that TAMs with an M2-like phenotype (markers CD163, CD204, and CD206) have a pro-tumor effect while M1-like TAMs (CD68, CD80, and CD86) may have an anti-tumor effect [13]. However, this simplified dichotomy may not hold for all primary tumor sites, as high CD68+ TAM density has been identified as a negative prognostic indicator in breast and gastric cancer [22, 23]. Similarly, our analysis identified high CD68+ TAM/tumor cell clustering as having a strong association with worse OS. Additionally, high CD163+ TAM cell density was associated with worse OS. Together, these findings suggest that high TAM infiltration may portend a worse prognosis in metastatic ccRCC regardless of M1/M2 polarization.

Additionally, this analysis identified that CD68+ TAMs tend to cluster with tumor cells and away from stromal cells, while CD163+ and CD206+ TAMs tend to cluster with stromal cells and away from tumor cells (Fig 3B). Interestingly, in a 2018 study using dynamic imaging microscopy to directly observe TAM and CD8+ T-cell interactions in squamous cell lung cancer, Peranzoni et al remarked as an aside that CD163+ and CD206+ TAMs could easily be found in the stroma, while TAMs in the tumor core seldom expressed these markers [24]. Our analysis notes a similar observation using quantifiable and reproducible metrics, potentially shedding light on a biologic affinity for M2 polarized TAMs to the stromal compartment of malignant tumors. Further biologic investigation of this association is needed.

Significant limitations to this analysis include the relatively small sample size (35 patients; 55 samples; 129 ROIs), and retrospective nature of the study. This relatively small population increases the risk of making Type-II errors in outcomes that were not reported as statistically significant. The post-hoc power analysis indicated a minimum detectable hazard ratio of 1.7 for a standard continuous variable in a univariable Cox proportional hazard regression analysis. As such, findings with hazard ratios below 1.7 are at risk for being false negative findings in this study. Additionally, these findings were not confirmed in a validation cohort, and certainly require prospective replication in larger cohorts. These findings certainly require prospective replication in larger cohorts. Additionally, this study did not investigate the potential biologic effects that TAMs exert on the TIME, and only assessed their spatial organization and heterogeneity. An inherent limitation to spatial analysis is that it is not feasible to reliably measure clustering when very few cells of interest are present in the ROI, resulting in several

patients in our cohort being excluded from the survival analysis as it related to the clustering of relatively rare cell markers such as CD163 and CD206.

## Conclusion

Overall, we identified the novel finding that CD68+ TAMs preferentially cluster into the tumor compartment at the tumor/stroma interface, that CD163+ and CD206+ TAMs preferentially cluster into the stromal compartment, and identified worse survival for metastatic ccRCC patients with increased spatial clustering of CD68+ TAMs and tumor cells. These findings highlight the importance of including spatial analysis in studies of immune infiltration of tumors.

## Supporting information

**S1 Fig. KM estimates and multivariable Cox regressions for OS, for CD163+ and CD206+ cell density using optimal cut-points (CD163+ = 130.349, CD206+ = 22.738), log-rank p values reported.**
(TIF)

**S1 Data.**
(CSV)

## Author Contributions

**Conceptualization:** Nicholas H. Chakiryan, Gregory J. Kimmel, Youngchul Kim, Ali Hajiran, Ahmet M. Aydin, Logan Zemp, Esther Katende, Jad Chahoud, Philippe E. Spiess, Michelle Fournier, Liang Wang, Asmaa El-Kenawi, James Mulé, Philipp M. Altrock, Brandon J. Manley.

**Data curation:** Nicholas H. Chakiryan, Ahmet M. Aydin, Logan Zemp, Esther Katende, Jonathan Nguyen, Neale Lopez-Blanco, Carlos Moran-Segura, Brandon J. Manley.

**Formal analysis:** Nicholas H. Chakiryan, Gregory J. Kimmel, Youngchul Kim, Jonathan Nguyen, Neale Lopez-Blanco, Liang Wang, Carlos Moran-Segura, Philipp M. Altrock, Brandon J. Manley.

**Funding acquisition:** Brandon J. Manley.

**Investigation:** Nicholas H. Chakiryan, Gregory J. Kimmel, Jonathan Nguyen, Jasreman Dhillon, Brandon J. Manley.

**Methodology:** Nicholas H. Chakiryan, Gregory J. Kimmel, Youngchul Kim, Ali Hajiran, Jonathan Nguyen, Neale Lopez-Blanco, Jasreman Dhillon, Carlos Moran-Segura, James Mulé, Philipp M. Altrock, Brandon J. Manley.

**Project administration:** Esther Katende.

**Software:** Jonathan Nguyen.

**Supervision:** James Mulé, Philipp M. Altrock, Brandon J. Manley.

**Visualization:** Nicholas H. Chakiryan, Ali Hajiran.

**Writing – original draft:** Nicholas H. Chakiryan, Brandon J. Manley.

**Writing – review & editing:** Ali Hajiran, Ahmet M. Aydin, Logan Zemp, Esther Katende, Jonathan Nguyen, Jad Chahoud, Philippe E. Spiess, Michelle Fournier, Jasreman Dhillon,

Liang Wang, Carlos Moran-Segura, Asmaa El-Kenawi, James Mulé, Philipp M. Altrock, Brandon J. Manley.

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
