## [Decision Letter · Decision Letter 0]

10 Feb 2021

PONE-D-20-39779

Spatial Clustering of CD68+ Tumor Associated Macrophages with Tumor Cells is Associated with Worse Overall Survival in Metastatic Clear Cell Renal Cell Carcinoma

PLOS ONE

Dear Dr. Chakiryan,

Thank you for submitting your manuscript to PLOS ONE. After careful consideration, we feel that it has merit but does not fully meet PLOS ONE’s publication criteria as it currently stands. Therefore, we invite you to submit a revised version of the manuscript that addresses the points raised during the review process.

We look forward to receiving your revised manuscript.

Kind regards,

Pankaj K Singh, Ph.D.

Academic Editor

PLOS ONE

Journal Requirements:

2. Please include your table as part of your main manuscript and remove the individual file. Please note that supplementary tables should be uploaded as separate "supporting information" files.

3. Thank you for including your ethics statement: 

"All tumor samples were obtained through protocols approved by the institutional review board(H.Lee Moffitt Cancer Center and Research Institute’s Total Cancer Care protocol MCC#14690; Advarra IRB Pro00014441).Written informed consents were obtained from all tissue donors".   

a. Please amend your current ethics statement to confirm that your named institutional review board or ethics committee specifically approved this study.

'CONFLICT OF INTEREST DISCLOSURE

The corresponding author certifies that all conflicts of interest, including specific financial interests and relationships and affiliations relevant to the subject matter or materials discussed in the manuscript (ie. employment/affiliation, grants or funding, consultancies, honoraria, stock ownership or options, expert testimony, royalties, or patents filed, received, or pending), are the following: NHC, GJK, AH, AMA, LZ, JN, JC, SF, MF, JD, SM, CM, EK, PMA, and YK have no disclosures; BJM is an NCCN Kidney Cancer Panel Member; PES is an NCCN Bladder and Penile Cancer Panel Member and Vice-Chair; JM is an Associate Center Director at Moffitt Cancer Center, has ownership interest in Fulgent Genetics, Inc., Aleta Biotherapeutics, Inc., Cold Genesys, Inc., Myst Pharma, Inc., and Tailored Therapeutics, Inc., and is a consultant/advisory board member for ONCoPEP, Inc., Cold Genesys, Inc., Morphogenesis, Inc., Mersana Therapeutics, Inc., GammaDelta Therapeutics, Ltd., Myst Pharma, Inc., Tailored Therapeutics, Inc., Verseau Therapeutics, Inc., Iovance Biotherapeutics, Inc., Vault Pharma, Inc., Noble Life Sciences Partners, Fulgent Genetics, Inc., UbiVac, LLC, Vycellix, Inc., and Aleta Biotherapeutics, Inc.'

a. Please confirm that this does not alter your adherence to all PLOS ONE policies on sharing data and materials, by including the following statement: "This does not alter our adherence to  PLOS ONE policies on sharing data and materials.” (as detailed online in our guide for authors http://journals.plos.org/plosone/s/competing-interests).  If there are restrictions on sharing of data and/or materials, please state these.

Please note that we cannot proceed with consideration of your article until this information has been declared.

6. Please amend your list of authors on the manuscript to ensure that each author is linked to an affiliation. Authors’ affiliations should reflect the institution where the work was done (if authors moved subsequently, you can also list the new affiliation stating “current affiliation:….” as necessary).

7. Please include a separate caption for each figure in your manuscript.

8. Please include captions for your Supporting Information files at the end of your manuscript, and update any in-text citations to match accordingly. Please see our Supporting Information guidelines for more information: http://journals.plos.org/plosone/s/supporting-information

Reviewers' comments:

Reviewer's Responses to Questions

**Comments to the Author**

1. Is the manuscript technically sound, and do the data support the conclusions?

Reviewer #1: Yes

Reviewer #2: Yes

2. Has the statistical analysis been performed appropriately and rigorously? 

Reviewer #1: No

Reviewer #2: Yes

3. Have the authors made all data underlying the findings in their manuscript fully available?

Reviewer #1: Yes

Reviewer #2: Yes

4. Is the manuscript presented in an intelligible fashion and written in standard English?

Reviewer #1: Yes

Reviewer #2: Yes

5. Review Comments to the Author

Reviewer #1: The authors conducted a study aiming to assess cellular density and cellular clustering for myeloid cell markers in 129 regions of interest from 55 samples from 35 patients with metastatic ccRCC. CD68+ density was not associated with OS, while high tumor/CD68+ cell clustering was associated with significantly worse OS. I have some concerns on the study design and statistical analysis.

1. The authors need to provide the sample size calculation and power analysis to justify the design of the study.

2. The definition of overall survival is starting from the sample collection time which may varies across patients. The starting time is suggested to use either diagnosis time or treatment starting time.

3. Have authors conducted assessment to test the Cox proportional hazards assumption?

4. The major limitation of the study is that, there is no validation cohort of the study, it needs to be discussed in the manuscript.

Reviewer #2: In this manuscript by Chakriyan et. al., the authors have performed spatial clustering analysis of macrophage markers in metastatic clear cell renal cell carcinoma samples using multiplex immunohistochemistry. The authors have analyzed spatial clusters of macrophage markers with tumor and stromal sections of the tumor tissue to predict the effect on overall survival. It is an interesting approach to analyze and predict overall survival in patients. As mentioned by authors the main limitation of the study is small sample size. The authors show that CD68 spatial distribution has a role to play in predicting OS. The CD163 and CD206 are M2 macrophage markers and are also associated with worse OS in multiple cancer studies. The experimental data would have been more informative and relevant if public database-based OS were compared with the multiplex immunohistochemistry-based spatial clustering of markers. Overall, the article presents an interesting approach to predict overall OS based on spatial distribution of markers. The authors should address the concerns listed:

1. The authors should discuss the relevance of CD163 and CD206 marker clustering in the stromal regions. Since tumor cells secrete a variety of factors that polarize macrophages to M2 type, it is intriguing to note why M2 markers are localized in stromal compartment.

2. Have the authors checked the spatial distribution of CD80 and CD86? Is the staining pattern distribution of these markers similar to CD68?

3. In the Figure 2, authors have stained the tissues for PD-L1 also. The authors should describe the PD-L1 expression in results.

4. Did the authors observe any difference in spatial distribution of CD68 in tissues when segregated based on gender and does it have any effect on OS also?

6. PLOS authors have the option to publish the peer review history of their article (what does this mean?). If published, this will include your full peer review and any attached files.

Reviewer #1: No

Reviewer #2: No

---

## [Author Response · Author response to Decision Letter 0]

26 Feb 2021

NOTE: The response to reviewers is also included as an attached file, which includes an appendix with supporting information to accompany the responses. 

RESPONSE TO REVIEWERS

We would like to thank the reviewers for their careful reading of the manuscript and thoughtful feedback. 

Reviewer #1: The authors conducted a study aiming to assess cellular density and cellular clustering for myeloid cell markers in 129 regions of interest from 55 samples from 35 patients with metastatic ccRCC. CD68+ density was not associated with OS, while high tumor/CD68+ cell clustering was associated with significantly worse OS. I have some concerns on the study design and statistical analysis.

1. The authors need to provide the sample size calculation and power analysis to justify the design of the study.

 This study was conceived as a pilot feasibility study to explore the effect of TAM/tumor spatial clustering on overall survival, a relationship that has not been previously assessed. As such, our study design was not based on an explicit statistical hypothesis but we were interested in discovering large effect sizes. A power/sample size analysis using a power of 0.80, and alpha of 0.05, with an 80% event rate indicates that the 35-patient cohort in our analysis would be adequate to detect a minimum detectable hazard ratio (MDHR) of 1.7 for a standardized continuous variable in a univariate Cox proportional hazards regression analysis. This has been included in the methods, results, and discussion sections of the manuscript (lines 143 to 144, and 181 to 184).

 We consider this MDHR to be appropriate, as we were interested in discovering large effect sizes. We certainly acknowledge that with a small cohort we are at risk of making Type-II errors in outcomes which were not reported as significant in the manuscript. This limitation has been added to the discussion section of the manuscript (lines 223 to 227).

2. The definition of overall survival is starting from the sample collection time which may varies across patients. The starting time is suggested to use either diagnosis time or treatment starting time.

 We explicitly chose OS from the time of sample collection when the study was designed. The specific variables of interest in this study were immune cell clustering and density, which are identified in a snapshot that occurs exactly at the time of sample collection. As such, this reflects survival from the moment in time that these variables were measured, which we feel is the appropriate timeframe. 

 Additionally, all samples included in this study were collected in the brief pre-treatment window from patients with metastatic ccRCC, and thus the time from diagnosis to sample collection is very short. The median time from diagnosis to sample collection was 26 days (IQR 9 – 42). 

3. Have authors conducted assessment to test the Cox proportional hazards assumption?

 We have included an assessment of the Cox proportional hazards assumption of the primary Cox regression model (Figure 3C). None of the individual covariates or the Schoenfeld’s global chi-square test violated the proportional hazards assumption. This is included at the bottom of this file as an Appendix.

4. The major limitation of the study is that, there is no validation cohort of the study, it needs to be discussed in the manuscript.

 We agree completely. This analysis certainly warrants prospective validation in a larger cohort. We have added this into the limitations section (lines 227 to 228).

Reviewer #2: In this manuscript by Chakriyan et. al., the authors have performed spatial clustering analysis of macrophage markers in metastatic clear cell renal cell carcinoma samples using multiplex immunohistochemistry. The authors have analyzed spatial clusters of macrophage markers with tumor and stromal sections of the tumor tissue to predict the effect on overall survival. It is an interesting approach to analyze and predict overall survival in patients. As mentioned by authors the main limitation of the study is small sample size. The authors show that CD68 spatial distribution has a role to play in predicting OS. The CD163 and CD206 are M2 macrophage markers and are also associated with worse OS in multiple cancer studies. The experimental data would have been more informative and relevant if public database-based OS were compared with the multiplex immunohistochemistry-based spatial clustering of markers. Overall, the article presents an interesting approach to predict overall OS based on spatial distribution of markers. The authors should address the concerns listed:

1. The authors should discuss the relevance of CD163 and CD206 marker clustering in the stromal regions. Since tumor cells secrete a variety of factors that polarize macrophages to M2 type, it is intriguing to note why M2 markers are localized in stromal compartment.

 We agree that this is a highly interesting finding. We do not have an answer as to why CD163+ and CD206+ TAMs preferentially cluster into the stromal compartment. However, it is worth noting that this relationship was remarked upon in a prior study, though it was not the purpose of that study and was not quantified (Reference #24 in the manuscript). Further work regarding macrophage chemotaxis into the tumor microenvironment is warranted. For our part, we will first pursue validation of these findings in a larger IF cohort, to confirm this association. 

2. Have the authors checked the spatial distribution of CD80 and CD86? Is the staining pattern distribution of these markers similar to CD68?

 This is an interesting question, but unfortunately the myeloid panel used in this study did not include the CD80 or CD86 markers. 

3. In the Figure 2, authors have stained the tissues for PD-L1 also. The authors should describe the PD-L1 expression in results.

 Thank you for noticing this discrepancy. We have included the PDL1 distribution results in the results section (lines 168 to 170). 

4. Did the authors observe any difference in spatial distribution of CD68 in tissues when segregated based on gender and does it have any effect on OS also?

 These are interesting questions. We have included this data below as an Appendix to this file. There was no difference in CD68/Tumor clustering or OS between genders in our cohort.

---

## [Decision Letter · Decision Letter 1]

6 Apr 2021

Spatial Clustering of CD68+ Tumor Associated Macrophages with Tumor Cells is Associated with Worse Overall Survival in Metastatic Clear Cell Renal Cell Carcinoma

PONE-D-20-39779R1

Dear Dr. Chakiryan,

We’re pleased to inform you that your manuscript has been judged scientifically suitable for publication and will be formally accepted for publication once it meets all outstanding technical requirements.

Kind regards,

Pankaj K Singh, Ph.D.

Academic Editor

PLOS ONE

Additional Editor Comments (optional):

Reviewers' comments:

Reviewer's Responses to Questions

**Comments to the Author**

1. If the authors have adequately addressed your comments raised in a previous round of review and you feel that this manuscript is now acceptable for publication, you may indicate that here to bypass the “Comments to the Author” section, enter your conflict of interest statement in the “Confidential to Editor” section, and submit your "Accept" recommendation.

Reviewer #1: All comments have been addressed

Reviewer #2: All comments have been addressed

2. Is the manuscript technically sound, and do the data support the conclusions?

Reviewer #1: Yes

Reviewer #2: Yes

3. Has the statistical analysis been performed appropriately and rigorously? 

Reviewer #1: Yes

Reviewer #2: Yes

4. Have the authors made all data underlying the findings in their manuscript fully available?

Reviewer #1: Yes

Reviewer #2: Yes

5. Is the manuscript presented in an intelligible fashion and written in standard English?

Reviewer #1: (No Response)

Reviewer #2: Yes

6. Review Comments to the Author

Reviewer #1: (No Response)

Reviewer #2: The authors have addressed all the comments satisfactorily by either performing the experiments or providing justifications.

7. PLOS authors have the option to publish the peer review history of their article (what does this mean?). If published, this will include your full peer review and any attached files.

Reviewer #1: **Yes: **Wei Xu

Reviewer #2: No

---

## [Editor Report · Acceptance letter]

8 Apr 2021

PONE-D-20-39779R1 

Spatial Clustering of CD68+ Tumor Associated Macrophages with Tumor Cells is Associated with Worse Overall Survival in Metastatic Clear Cell Renal Cell Carcinoma 

Dear Dr. Chakiryan:

I'm pleased to inform you that your manuscript has been deemed suitable for publication in PLOS ONE. Congratulations! Your manuscript is now with our production department. 

Kind regards, 

on behalf of

Dr. Pankaj K Singh 

Academic Editor

PLOS ONE